# The Formation of Chitosan-Coated Rhamnolipid Liposomes Containing Curcumin: Stability and In Vitro Digestion

**DOI:** 10.3390/molecules26030560

**Published:** 2021-01-21

**Authors:** Wei Zhou, Ce Cheng, Li Ma, Liqiang Zou, Wei Liu, Ruyi Li, Yupo Cao, Yuhuan Liu, Roger Ruan, Jihua Li

**Affiliations:** 1State Key Laboratory of Food Science and Technology, Nanchang University, Nanchang 330047, Jiangxi, China; weizhou111@foxmail.com (W.Z.); 352313318015@email.ncu.edu.cn (C.C.); 402313318047@email.ncu.edu.cn (L.M.); zouliqiang2010@163.com (L.Z.); liuyuhuan@ncu.edu.cn (Y.L.); ruanx001@umn.edu (R.R.); 2Key Laboratory of Tropical Crop Products Processing of Ministry of Agriculture and Rural Affairs, Agricultural Products Processing Research Institute, Chinese Academy of Tropical Agricultural Sciences, Zhanjiang 524001, Guangdong, China; yupo53@163.com (Y.C.); foodpaper@126.com (J.L.); 3National R&D Center for Freshwater Fish Processing, Jiangxi Normal University, Nanchang 330022, Jiangxi, China; 4Hainan Key Laboratory of Storage and Processing of Fruits and Vegetables, Zhanjiang 524001, Guangdong, China

**Keywords:** liposomes, chitosan, curcumin, rhamnolipids, stability, in vitro digestion

## Abstract

There is growing interest in developing biomaterial-coated liposome delivery systems to improve the stability and bioavailability of curcumin, which is a hydrophobic nutraceutical claimed to have several health benefits. The curcumin-loaded rhamnolipid liposomes (Cur-RL-Lips) were fabricated from rhamnolipid and phospholipids, and then chitosan (CS) covered the surface of Cur-RL-Lips by electrostatic interaction to form CS-coated Cur-RL-Lips. The influence of CS concentration on the physical stability and digestion of the liposomes was investigated. The CS-coated Cur-RL-Lips with RL:CS = 1:1 have a relatively small size (412.9 nm) and positive charge (19.7 mV). The CS-coated Cur-RL-Lips remained stable from pH 2 to 5 at room temperature and can effectively slow the degradation of curcumin at 80 °C; however, they were highly unstable to salt addition. In addition, compared with Cur-RL-Lips, the bioavailability of curcumin in CS-coated Cur-RL-Lips was relatively high due to its high transformation in gastrointestinal tract. These results may facilitate the design of a more efficacious liposomal delivery system that enhances the stability and bioavailability of curcumin in nutraceutical-loaded functional foods and beverages.

## 1. Introduction

Liposomes (Lips) are spherically shaped vesicles consisting of a phospholipid bilayer that surrounds an aqueous core; they have attracted widespread attention for their good nutraceutical-loaded property [1]. Due to their phospholipid bilayer, liposomes are often used for encapsulating both hydrophilic and lipophilic functional ingredients and controlling their release [2,3]. Therefore, they have considerable potential for application in various fields such as functional foods, nutraceutical supplements, and pharmaceuticals. However, the application of conventional liposomes was limited because of their poor stability, their susceptibility to acidic pH, oxidation and enzymolysis with subsequent possibility of losing the loaded bioactive agents [4,5,6,7]. Currently, a number of researchers have proposed liposomal surface modifications as a way to overcome these limitations. Various polymers were successfully applied to improve the surface properties of liposomes, including synthetic polymers (pluronics and polyethylene glycol) [8,9,10] and natural polymer (peptide, protein, and polysaccharides) [11,12]. Among them, polysaccharides are the most extensively used polymers for liposome modification owing to non-toxicity, non-immunogenicity, good biocompatibility and biodegradability. Hence, it is necessary to select suitable polysaccharides to modify liposomes.

Chitosan (CS) is obtained by partial deacetylation of chitin from crustacean shells, which consists of d-glucosamine and *N*-acetyl glucosamine units linked through a β (1 → 4) linkage [11]. When CS is dissolved in acidic media, CS has cationic charge due to protonation of the -NH_3_ function on repeating the d-glucosamine unit [13]. Based on electrostatic interactions, cationic CS can easily attach to the surface of anionic macromolecules and form a protective polyelectrolyte layer [4]. Additionally, CS can improve the penetration of macromolecules due to its mucoadhesive properties, and this property has risen wide-spread attention as a potential enhancer of absorption across the mucosal epithelia [11]. Furthermore, several studies have demonstrated that CS may increase the cellular permeability [14]. Take these properties of CS into account, it can be used for a coating material to cover the surface of anion liposomes by the electrostatic interaction, and thus improve the permeability and stability of liposomes.

In this study, curcumin (Cur) was utilized as a model bioactive substance due to its good activity, such as antioxidant, anticancer, antibacterial, and anti-inflammatory [15]. The use of curcuminoid is, however, extremely limited, owing to its poor water-solubility, easy chemical degradation, and low bioavailability [16]. In order to overcome these disadvantages of curcuminoid, some researchers tried various kinds of delivery systems to load, protect, and deliver it, containing emulsion [16], nanoparticles [17], liposomes [18,19], and hydrogels [20]. In a previous study, our research groups successfully prepared curcumin-loaded rhamnolipid liposomes (Cur-RL-Lips) and characterized their physicochemical properties [21]. However, the capacity of Cur-RL-Lips was limited to inhibit the temperature-induced curcumin degradation. Thus, we selected CS as a modifier to produce curcumin-loaded liposomal delivery systems for improving the stability and bioavailability of curcumin. The influence of the CS level on the stability, gastrointestinal fate, and curcuminoid bioavailability of Cur-loaded liposomes was investigated. This study should provide useful insights into the design of a more efficacious liposomal delivery system that enhances the stability and bioavailability of curcumin in nutraceutical-loaded functional foods and beverages.

## 2. Results and Discussion

### 2.1. Formation of CS-Coated Cur-RL-Lips

The formation of CS-coated Cur-RL-Lips was mainly attributed to the electrostatic interaction between CS with positive charge and Cur-RL-Lips with negative charge, as schematized in Figure 1a. The visual appearance, mean diameter and ξ-potential of liposomes suspensions with different ratios of RL/CS are shown in Figure 1b,c, respectively. At a fixed Cur-RL-Lips concentration, the turbidity of the liposome suspensions increased with increasing CS level (Figure 1b). The mean diameter and ξ-potential of Cur-RL-Lips (control) were 228.7 nm and −14.5 mV, respectively. However, the cationic CS solution was added to anionic Cur-RL-Lips suspensions, resulting in the change of mean diameter and ξ-potential (Figure 1c). The mean diameter of the liposome suspensions increased with increasing the ratios of RL/CS: mean diameter = 412.9, 541.3, and 639.8 nm for liposomes with RL:CS mass ratios of 1:1, 1:2, and 1:3, respectively. The increase in particle size may have occurred for two reasons: (i) the addition of CS in the Cur-RL-Lips resulted in the aggregation of liposomes; (ii) CS covered the surface of Cur-RL-Lips, which caused the thickness of the liposomes’ shells to increase. Meanwhile, the ξ-potential of the liposome suspensions changed from −14.5 mV (Cur-RL-Lips) to 19.7 mV (CS-coated Cur-RL-Lips with RL:CS =1:1) and has no obvious difference with increasing CS level. The chitosan with positive charge can form a protective polyelectrostatic layer onto liposomes with negative charge, which was based on the electrostatic interaction between positive charge and negative charge [22]. In this situation, the negative charge of the Cur-RL-Lips surface has been fully shielded by the CS with positive charge. As the Cur-RL-Lips surface becomes fully covered with CS, the surface charge of the liposomes becomes positive and the repulsion among the CS-coated liposomes dominates [3].

Additionally, the microstructure of the liposomes was obtained using AFM topographic images (Figure 2). The Cur-RL-Lips (control) and CS-coated Cur-RL-Lips with different ratios of RL/CS (RL:CS =1:1 and 1:2) appeared small in particle size and had good dispersivity. However, the CS-coated Cur-RL-Lips with RL:CS = 1:3 showed clearly the aggregation of liposomes and low dispersivity, which suggested that excessive CS can break the stability of liposomes. The result was consistent with those measured by dynamic light scattering (Figure 1). The heights of CS-coated Cur-RL-Lips were higher than those of Cur-RL-Lips, indicating that CS was successfully coated on the surface of Cur-RL-Lips. Previous studies have also shown that CS-modified liposomes have bigger sizes and more rigidity than non-modified liposomes, which can influence particle stability, digestion and the release property of the loaded drug [23,24].

### 2.2. Stability of CS-Coated Cur-RL-Lips

The liposomes could experience various environmental stresses when they were added to commercial products (such as foodstuffs, nutritional supplements, and medicines) or when they travel through the human gastrointestinal tract (GIT) [19,25]. These different environmental conditions (e.g., pH value, salt level, and temperature) could affect the stability of liposomes. Thus, this is quite necessary to evaluate the influence of environmental factors on the stability of liposomes.

#### 2.2.1. pH Stability

Liposome suspensions were exposed to various pH values (pH 2~7) for 3 days, and then their visual appearance, particle size, and surface charge were measured. As shown in Figure 3a, fresh Cur-RL-Lips (control) suspensions were a yellow, clear and transparent liquid at different pH values. After storage for 3 days, there was no significant change in the visual appearance of Cur-RL-Lips suspensions. However, CS-coated Cur-RL-Lips suspensions with different ratios of RL/CS were yellow cloudy liquid. The turbidity of CS-coated Cur-RL-Lips suspensions was the highest at pH 7, while their turbidity decreased with reducing pH values from 7 to 2. Visually, a sediment was observed at the bottom of the test bottles under nearly neutral pH (6 and 7) after 3 days of storage, indicating some aggregation of liposomes had occurred. Conversely, CS-coated Cur-RL-Lips were relatively stable at the most acidic solutions (pH 2 and 3), which is conducive to endure the acidic condition of stomach and prevent the release of the encapsulated drug in the stomach [26,27]. These phenomena were supported by dynamic light scattering. At pH 7, except for Cur-RL-Lips (control), there was a marked increase in the particle size of CS-coated liposomes after 3-day storage (Figure 3b). The CS-coated liposomes with RL:CS = 1:1 had significant increase in particle size at pH 4.5 for 3-day storage, others were not significantly different at pH 2 and 4.5. Additionally, compared with liposomes at pH 4.5, the particle size of liposomes was small at pH 2, which may be due to a good solubility of CS at acidic pH [28]. To provide some further insights into the mechanism of liposomes instability, the surface charge of liposomes was determined at various pH values (Figure 3c). The surface charge of Cur-RL-Lips (control) went from highly negative at neutral pH (−37.9 mV for pH 7) to highly positive at acidic pH (18.0 mV for pH 2), with a zero charge at around pH 4. Meanwhile, the change of surface charge has no effect on the stability of Cur-RL-Lips. However, CS-coated Cur-RL-Lips with different ratios of RL/CS have a positive surface charge, and the magnitude of surface charge was the lowest at pH 7. Therefore, there are two reasons for explaining the instability of CS-coated Cur-RL-Lips at pH 6 and 7: (i) the electrostatic repulsion between liposomes was low because of the low magnitude of surface charge at pH 7; (ii) the chitosan has low solubility at an alkaline and neutral pH so that the CS-coated liposomes precipitate out [29].

#### 2.2.2. Salt Stability

The salt stability of the liposomes was determined by adding different levels of NaCl (0~400 mM) to them (Figure 4). The turbidity of all samples increased clearly when adding salt (Figure 4a). After 3-day storage, Cur-RL-Lips suspensions had only a slight change in the visual appearance and particle size (Figure 4a,b), though the added salt resulted in a decrease in the magnitude of negative charge on the Cur-RL-Lips (Figure 4c). Instead, CS-coated Cur-RL-Lips were extremely unstable for adding salt. After 3-day storage, visual appearance showed that a yellow sediment formed at the bottom of test bottles, especially CS-coated Cur-RL-Lips with RL:CS = 1:2 and 1:3, and particle size increased strongly as the level of salt added was increased. These results indicate that the aggregation of CS-coated liposomes has occurred, which could be because of the electrostatic screening between them by the mineral ions [30]. The magnitude of positive surface charge on the CS-coated liposomes reduced with increasing salt level, which again supported the viewpoint of the electrostatic screening. Overall, these results suggest that it is adverse to the use of CS-coated Cur-RL-Lips under the high salt condition. 

#### 2.2.3. Thermal Stability

The capacity of nutraceutical-loaded liposomes to remain stable during high temperature (e.g., cooking, sterilization, or pasteurization) is important for their commercial applications. Hence, the thermal stability of Cur-loaded liposomes was determined by measuring the curcumin degradation and the change of particle size at 80 °C for 60 min (Figure 5). The concentration of curcumin in all samples decreased during storage. The degradation rate of curcumin in Cur-RL-Lips was the highest, which is probably because the structure of the thin-phospholipid bilayer cannot stop curcumin from chemical degradation during heating [31]. The degradation rate of curcumin in CS-coated liposomes slowed down when the RL/CS ratio increased (Figure 5a). The curcumin retention in the Cur-RL-Lips and CS-coated Cur-RL-Lips with RL:CS = 1:3 after storage for 60 min at 80 °C were 78.8% and 94.4%, respectively. These results indicated that CS covering on the surface of liposomes can effectively prevent curcumin from chemical degradation during thermal treatment. Additionally, except for CS-coated Cur-RL-Lips with RL:CS = 1:3, there was no obviously difference (*P* < 0.05) in the particle size of liposomes both before and after storage (Figure 5b). Therefore, the CS-coated Cur-RL-Lips are effective at protecting curcumin from chemical degradation, as well as protecting liposomes from disruption or aggregation as RL:CS ≤ 1:2.

### 2.3. Release of Curcumin in CS-Coated Cur-RL-Lips

The in vitro release behavior of curcumin was recorded during 72 h at 37 °C, and the cumulative release curve is shown in Figure 6. The Cur-RL-Lips showed a higher curcumin release than the CS-coated Cur-RL-Lips, and its cumulative release percentage reached 82.0% after 72 h. However, the addition of CS in Cur-loaded liposomes can result in the reduction in curcumin release. The curcumin release was only 67.6% from CS-coated Cur-RL-Lips with RL:CS = 1:3 in 72 h at 37 °C. These indicated that the presence of CS was able to inhibit the release of curcumin from the liposomes, which may be beneficial for prolonged release applications. Presumably, the surface of Cur-RL-Lips was fully covered with CS because of the electrostatic interaction between CS (positive charge) and Cur-RL-Lips (negative charge), thereby reducing the tendency of curcumin to be released [31].

### 2.4. In Vitro Digestion of CS-Coated Cur-RL-Lips

The digestion process of the liposomes was studied by a three-stage simulated GIT model, including mouth stage, stomach stage, and small intestine stage. At the end of each single stage, the visual appearance, particle size, and ζ-potential were recorded (Figure 7a–c). When the liposomes were exposed to the mouth phase, the particle size of all the liposomes (mean diameter > 1 μm) was obviously larger than that of the initial liposomes (mean diameter < 600 nm). Meanwhile, there was no change in surface charge of Cur-RL-Lips. However, the surface charge of CS-coated Cur-RL-Lips shifted positive to negative, and the magnitude of the surface charge decreased with increasing ratios of RL/CS in the mouth phase. These changes mainly attributed to the ability of mucin in simulated saliva to promote aggregation of liposomes and the effect of pH in surface charge of liposomes [32]. Next, the liposomes entered the stomach for digestion. Except for the particle size of CS-coated Cur-RL-Lips with RL:CS = 1:3 decreasing slightly, the particle size of the liposomes did not significantly change. Nevertheless, the surface charge of all liposomes changed from negative in the mouth phase to positive in the stomach phase, which was mainly due to the low pH of the gastric fluids [33]. In addition, the magnitude of the surface charge in CS-coated Cur-RL-Lips with RL:CS = 1:2 and 1:3 had an apparent increase alongside an increase in the electrostatic repulsion between the liposomes, which may be the main reason for reducing the particle size of CS-coated Cur-RL-Lips with RL:CS = 1:3. All samples had relatively large particle sizes and low negative surface charges after passing through the small intestine phase. An obvious sediment was shown in the bottom of test bottles, especially in Cur-RL-Lips. Furthermore, the particle size of samples with CS was smaller than those without CS, which indicated that the presence of CS may have inhibited the formation of large particle structures such as vesicles and insoluble calcium soaps. 

Additionally, the in vitro bioavailability of curcumin was determined by measuring initial curcumin concentration (*C_initial_*), as well as the curcumin concentration in raw digesta phase (*C_digesta_*) and in mixed micelle fraction (*C_micelle_*) at the end of small intestine stage. Then, these data were used to calculate the transformation (T*) and bioaccessibility (B*), as the following equations:(1)$$Transformation\;\left(\%\right)=\frac{C_{digesta}}{C_{initial}}\times 100$$
(2)$$Bioaccessibility\;\left(\%\right)=\frac{C_{micelle}}{C_{digesta}}\times 100$$

The bioavailability (BA) is given by B* × T*, so it is a ratio of the curcumin concentration in mixed micelle fraction to the initial curcumin concentration. As shown in Figure 7d, the CS-coated Cur-RL-Lips have higher transformation than Cur-RL-Lips, which indicated that the digesta of CS-coated Cur-RL-Lips remained at a higher concentration of curcumin. The bioaccessibility of curcumin was similar in liposomes, apart from CS-coated Cur-RL-Lips with RL:CS = 1:3. A high level of CS may cause by the aggregation of colloid, and thus affect the absorption of nutraceutical [34]. However, considering the bioavailability, the amount of bioavailable curcumin was higher in CS-coated Cur-RL-Lips than in Cur-RL-Lips, which is consistent with the study of Cuomo et al. [4]. This effect can mainly be attributed to the much higher transformation of the curcumin in the CS-coated Cur-RL-Lips. Consequently, these results indicated that a larger amount of curcumin can be absorbed by delivering it in the form of CS-coated liposomes, which is mainly because of a large increase in transformation. 

## 3. Materials and Methods

### 3.1. Materials

Phospholipid S100 (containing 94% phosphatidylcholine) was purchased from Lipoid GmbH (Ludwigshafen, Germany). Curcumin (98%) was purchased from Aladdin Industrial Corporation (Shanghai, China). Chitosan was purchased from Sigma-Aldrich Co. (St. Louis, MO, USA). Rhamnolipids were provided by Xi’an boliante Chemical Co., (Xi’an, China). All other reagents (analytical grade) were purchased from Xilong Chemical Co., (Shanghai, China). Double distilled water (Milli-Q) was used to prepare all solutions.

### 3.2. Preparation of Cur-RL-Lips and CS-Coated Cur-RL-Lips

The curcumin-loaded rhamnolipid liposomes (Cur-RL-Lips) were prepared according to the method of Cheng et al. [21] with some modifications. Phospholipids (RL, 10 mg/mL) were mixed with rhamnolipids (RL, 2 mg/mL) and a certain amount of curcumin (Cur) dissolved in absolute ethanol. The obtained solution was then added to a phosphate buffer saline solution (5 mM, pH 7) and stirred for 30 min. Aqueous Cur-RL-Lips was obtained by removing the ethanol using a vacuum rotary evaporator at 45 °C. The final curcumin concentration in this system was 400 μg/mL. 

The Cur-RL-Lips were coated with chitosan using the method of Cuomo et al. [4] with some modifications. The Cur-RL-Lips were mixed with equal volume chitosan (CS) solution to obtain a range of RL:CS mass ratios (1:1, 1:2 and 1:3 *w*/*w*). Finally, the Cur-RL-Lips (control) and CS-coated Cur-RL-Lips with different RL:CS mass ratios were adjusted to pH 4.5 by adding a small amount of either HCl or NaOH solution.

### 3.3. Mean Particle Diameter and ζ-Potential

The mean diameter and ζ-potential of the liposomes were determined using an instrument capable of both dynamic light scattering and microelectrophoresis analysis (Malvern Zetasizer Nano- ZSP, Malvern Instruments, Worcestershire, UK) according to the method of Peng et al. [18]. The liposome samples were diluted to an appropriate concentration using phosphate buffer solution (same pH and salt level as sample).

### 3.4. Microstructure

The microstructure of liposomes was observed using an atomic force microscope (AFM). The liposome suspensions were dropped into the clear mica plate, and then the microstructure images of liposomes were obtained by an AFM (Agilent 5500, Agilent Technologies, Santa Clara, CA, USA) with a silicon cantilever of force constant of 0.58 N/m operated in tapping mode at room temperature.

### 3.5. Stability of Cur-RL-Lips and CS-Coated Cur-RL-Lips

*pH*: The influence of pH on the stability of liposome dispersions was determined by preparing a series of samples with the desired pH values ranging from 2 to 7. The obtained samples were then stored in a 25 °C incubator for around 24 h and 3 d, respectively. The appearance, mean particle diameter, and ζ-potential of the samples were recorded using the methods described earlier.

*Salt addition*: The influence of salt level on the stability of liposome suspensions was determined by preparing a series of samples with different levels of NaCl (0, 50, 100, 200, and 400 mM). The obtained samples were then stored in a 25 °C incubator for around 24 h and 3 d, respectively. The appearance, mean particle diameter, and ζ-potential of the samples were recorded using the methods described earlier.

*Thermal stability*: The liposome suspensions (5 mL) were mixed with 20 mL of phosphate buffer solution (5 mM, pH 7) and sealed into a series of 2 mL test tubes, which were incubated in a water bath at 80 °C for 10, 20, 30, 40, 50, and 60 min, respectively. The tubes were then immediately cooled to room temperature using an ice water. Finally, the mean particle diameter was determined using the methods described earlier and the curcumin concentration was obtained by measuring the absorbance of samples at 420 nm.

### 3.6. Release of Curcumin in Cur-RL-Lips and CS-Coated Cur-RL-Lips

The in vitro release of curcumin from liposomes was evaluated according to the method of Li et al. [35]. Briefly, liposomal samples (2 mL) were transferred in dialysis bag (8–14 kDa molecular weight cut-off). The dialysis bags were then incubated in 50 mL of a phosphate buffer (5 mM, pH 7) release medium containing Tween-80 (0.5% *w*/*w*) and 10% ethanol as an auxiliary solvent for curcumin and incubated in a shaker at 100 rpm and at 37 °C for 72 h. At specific time intervals, 2 mL of release medium was withdrawn and replaced with fresh medium. The release medium was diluted with anhydrous ethanol to calculate the total cumulative amount of curcumin released from the liposomes. The content of curcumin was measured at various sampling times by a spectrophotometer at 420 nm.

The cumulative release (%) of curcumin was calculated using the equation below:(3)$$Cumulative\;release\;\left(\%\right)=(50C_{i}+\overset{n}{\underset{i=1}{\sum }}C_{i-1})V_{0}/M_{0}$$
where *V_0_* is the initial volume of curcumin in release system, *M_0_* is the initial mass of curcumin in the sample, and *C_i_* is the concentration of curcumin released at each sampling time point.

### 3.7. Digestion of Cur-RL-Lips and CS-Coated Cur-RL-Lips

A three-stage simulated GIT model was employed to characterize the potential gastrointestinal behavior of the liposomes [33,36]. (i) Mouth phase: the liposomes (7.5 mL) were combined with artificial saliva (7.5 mL), adjusted to pH 6.8, and then incubated at 37 °C for 2 min; (ii) Stomach phase: simulated gastric fluids (15 mL) containing 3.2 mg/L pepsin were added to the mouth phase (15 mL), the mixture was adjusted to pH 2.5, and then incubated at 37 °C for 2 h; (iii) Small intestine phase: the mixture (30 mL) from the stomach phase was collected and adjusted to pH 7.0, then added to 1.5 mL of simulated intestinal fluids and 3.5 mL bile salt solution. Afterward, the pH was adjusted back to 7.0, and 2.5 mL of pancreatin solution (36 mg/mL) was added. The temperature of the simulated gastrointestinal system was controlled at 37 °C throughout. The pH-stat method was used to monitor lipolysis during small intestinal digestion for 2 h.

### 3.8. Statistical Analysis

All experiments were performed on three prepared samples. The results were expressed as mean values ± standard deviation in triplicates. Statistical analysis software (SPSS 25.0) was used for analyzing the data. Analysis of variance (ANOVA) was used to determine significance at *p* < 0.05 by Tukey’s HSD (honestly significant difference) test.

## 4. Conclusions

In this study, the impact of the ratios of chitosan (CS) and rhamnolipid (RL) on the physical stability and digestion of the chitosan-coated rhamnolipid liposomes containing curcumin (CS-coated Cur-RL-Lips) was examined. Initially, we showed that chitosan (CS) and Cur-RL-Lips spontaneously assembled into CS-coated Cur-RL-Lips in aqueous solutions because of electrostatics interaction between cationic CS and anionic Cur-RL-Lips. Then, the physical stability and gastrointestinal digestion of CS-coated Cur-RL-Lips were investigated. The CS-coated Cur-RL-Lips were stable to aggregation under acidic conditions (pH ≤ 5), which is conducive to enduring the acidic condition of stomach and preventing the release of the encapsulated drug in the stomach. The CS-coated Cur-RL-Lips exhibited better stability than the Cur-RL-Lips when stored at elevated temperatures (80 °C for 60 min). The CS covered the surface of liposomes effectively inhibited temperature-induced curcumin degradation. Additionally, curcumin in CS-coated Cur-RL-Lips can be better absorbed than those in Cur-RL-Lips, but a high level of CS reduced the bioaccessibility of curcumin. Therefore, the amount of CS in liposomes should be controlled reasonably. Overall, these results have good potential for the design of a more efficacious liposomal delivery system for encapsulating and delivering bioactive agents. 

## Figures and Tables

**Figure 1 molecules-26-00560-f001:**
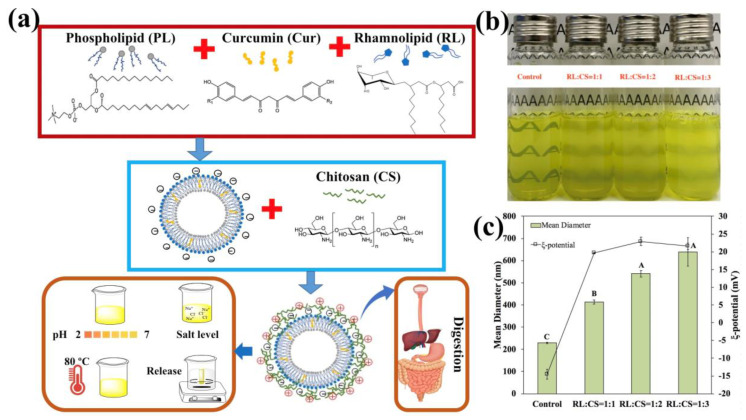
(**a**) The formation scheme of chitosan (CS)-coated curcumin-loaded rhamnolipid liposomes (Cur-RL-Lips); (**b**) visual appearance of liposomes with different ratios of RL/CS (RL:CS =1:1, 1:2, and 1:3); (**c**) mean diameter and ζ-potential of liposomes with different ratios of RL/CS. Samples denoted by different capital case letters (A–C) were significantly different (*p* < 0.05) when compared between different RL/CS ratios.

**Figure 2 molecules-26-00560-f002:**
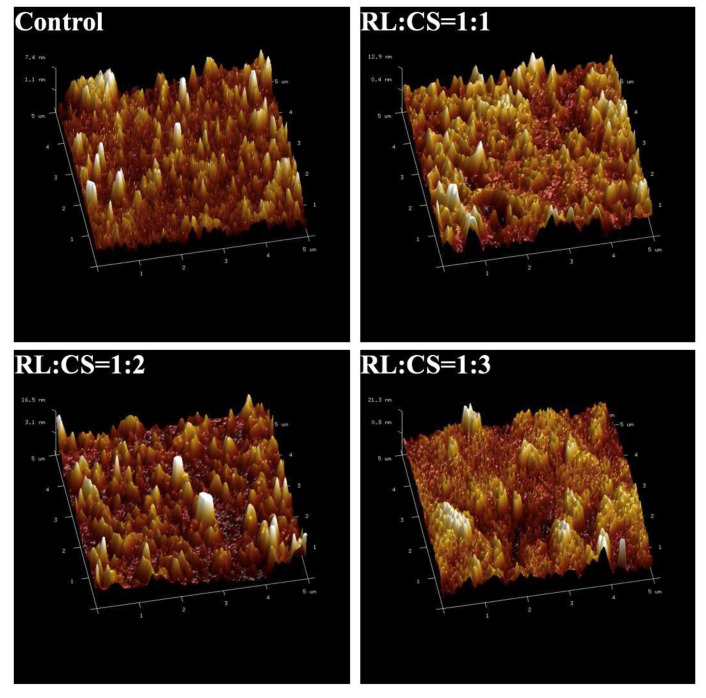
Atomic force microscopy image of Cur-RL-Lips (Control) and CS-coated Cur-RL-Lips with different ratios of RL/CS (RL:CS = 1:1, 1:2, and 1:3).

**Figure 3 molecules-26-00560-f003:**
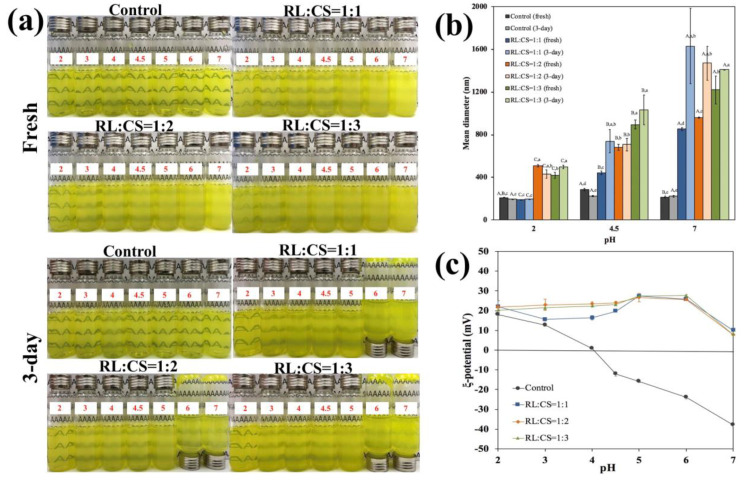
Influence of pH on the visual appearance (**a**), mean diameter (**b**) and ζ-potential (**c**) of liposomes with different ratios of RL/CS. Samples denoted by different capital case letters (A–C) were significantly different (*p* < 0.05) when compared between different pH (same RL/CS ratios and storage time). Samples denoted by lowercase letters (a–c) were significantly different (*p* < 0.05) when compared between different RL/CS ratios and storage times (same pH).

**Figure 4 molecules-26-00560-f004:**
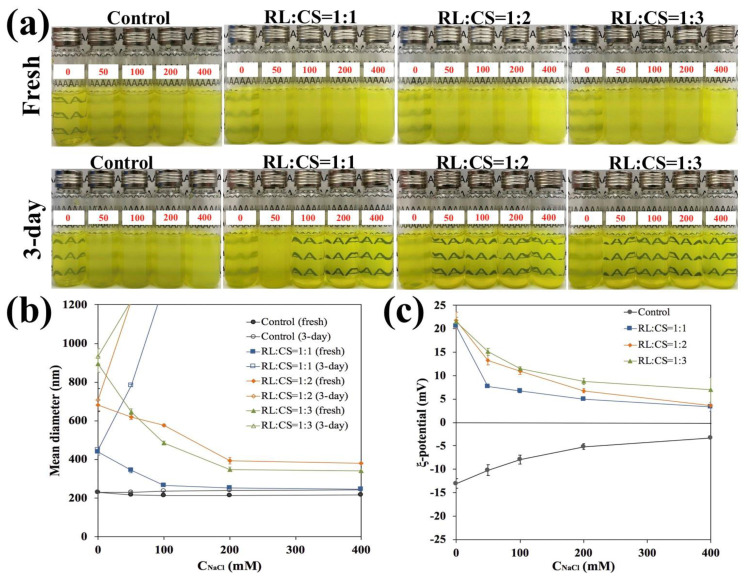
Influence of salt level on the visual appearance (**a**), mean diameter (**b**) and ζ-potential (**c**) of liposomes with different ratios of RL/CS.

**Figure 5 molecules-26-00560-f005:**
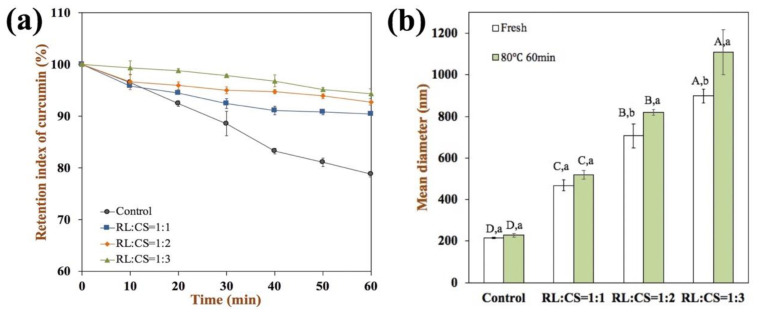
Degradation (**a**) and mean diameter (**b**) of Cur-loaded liposomes with different ratios of RL/CS at 80 °C for 60 min. Samples denoted by different capital case letters (A–C) were significantly different (*p* < 0.05) when compared between different RL/CS ratios. Samples denoted by lowercase letters (a–c) were significantly different (*p* < 0.05) when compared between different treatment condition.

**Figure 6 molecules-26-00560-f006:**
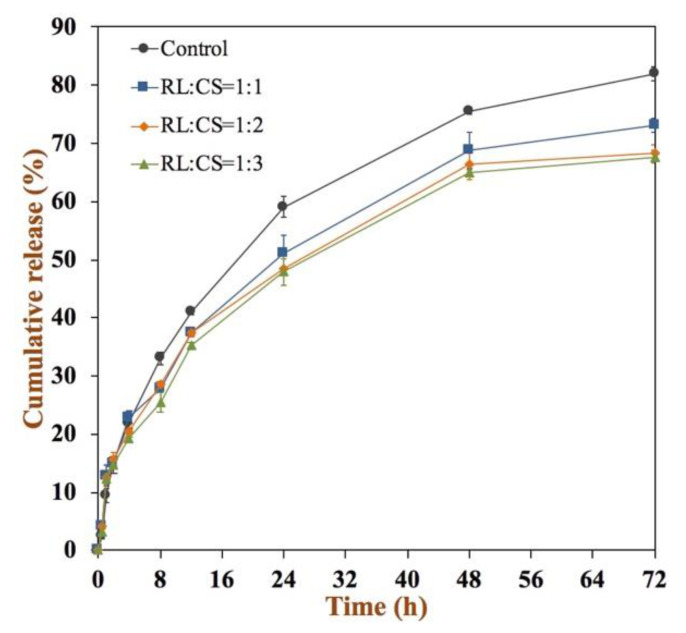
In vitro release curves of curcumin in Cur-loaded liposomes with different ratios of RL/CS at 37 °C for 72 h.

**Figure 7 molecules-26-00560-f007:**
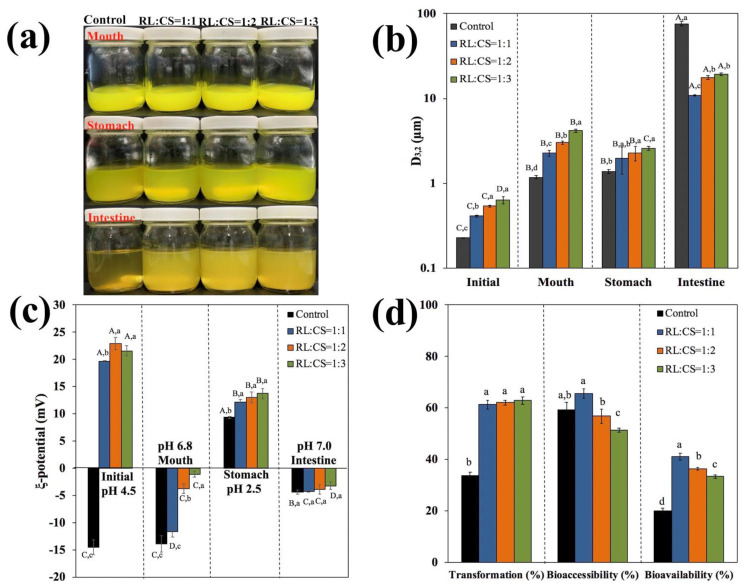
Visual appearance (**a**), particle size (**b**), ζ-potential (**c**) of Cur-loaded liposomes with different ratios of RL/CS during the digestion phases; (**d**) curcumin transformation, bioaccessibility and bioavailability percentages for Cur-loaded liposomes with different ratios of RL/CS during the digestion phases. Samples denoted by different capital case letters (A–D) were significantly different (*p* < 0.05) when compared between different GIT regions (same RL/CS ratios). Samples denoted by lowercase letters (a–c) were significantly different (*p* < 0.05) when compared between different RL/CS ratios (same GIT region).

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
