# Peer review of "The Formation of Chitosan-Coated Rhamnolipid Liposomes Containing Curcumin: Stability and In Vitro Digestion"

_molecules, 2021, doi:10.3390/molecules26030560_

Round 1

Reviewer 1 Report

  1. The originality of the work should be given in the introduction part
  2. It is hard to understand the molecular interaction as there is not any specific Spectroscopic evidence given in the manuscript

Author Response

1. The originality of the work should be given in the introduction part

Response: We thank the reviewer for their positive comments.  As recommended, we have revised the introduction to highlight the potential importance of upgrading our previous workand their potential application in the delivery of bioactive substances. Please see Lines 68-76.

2. It is hard to understand the molecular interaction as there is not any specific Spectroscopic evidence given in the manuscript

Response: We thank the reviewer for their constructive comments. In this study, the electrostatic interaction of the chitosan-coated rhamnolipid-liposomes containing curcumin (CS-coated Cur-RL-Lips) was determined by the change of ξ-potential on liposomes [Figure 1 (c)]. Please see Lines 91-98.

Reviewer 2 Report

The Manuscript ID foods-1064061 entitled “The formation of chitosan-coated rhamnolipid-liposomes containing curcumin: Stability and In vitro digestion” deals with the evaluation of the physical stability (pH, salts, temperature exposure) and bioavailability (in-vitro digestion) of curcumin as loaded into rhamnolipid-liposomes coated with chitosan. Such modified liposomes are believed of interest to produce and optimize curcumin delivery systems in foodstuffs, nutritional supplements and medicine.

In my opinion the topic is very interesting in the field of food science and technology, nutraceuticals and medicine.

The experimental work was just valuable mainly for the following reason. The work is a short upgrade of a previous work from the authors in which they evaluated the stability and availability of curcumin loaded in the same liposomes without chitosan coating. No attempts were made to evaluate the efficacy of such delivery system for example under conditions really simulating those experienced by the food materials during production of foods and beverages. In the present study, the physical stability of the liposomes was evaluated under a unique steady-state environmental condition, i.e. 80°C for 60 min. Unfortunately, the unit operations that are used in the food production involve transient conditions both in heat and mass transport, and results could be very different in terms of both in the physical stability of the liposomes and in curcumin bioavailability.

Some editing improvements are required. The equations are missing. The quality of the figures must be improved.

Line 302 : change mothed with method

Line 331-332 what do you mean?

Author Response

1.  The experimental work was just valuable mainly for the following reason. The work is a short upgrade of a previous work from the authors in which they evaluated the stability and availability of curcumin loaded in the same liposomes without chitosan coating. No attempts were made to evaluate the efficacy of such delivery system for example under conditions really simulating those experienced by the food materials during production of foods and beverages. In the present study, the physical stability of the liposomes was evaluated under a unique steady-state environmental condition, i.e. 80 °Cfor 60 min. Unfortunately, the unit operations that are used in the food production involve transient conditions both in heat and mass transport, and results could be very different in terms of both in the physical stability of the liposomes and in curcumin bioavailability.

Response: We thank the reviewer for their suggestion. In order to determine the stability of new curcumin-liposomes systems in food production (e.g. high temperature sterilization), we selected the simulation condition (80ºC for 60 min) to measure the thermal stability of liposomes and referred to the method of Peng et al.[1] Additionally, the release and bioavailability of curcumin  in Cur-loaded liposomes were determined by the experiments of in vivo release and digestion at 37 ºC, which can relatively truly reflect the release and bioavailability of curcumin.

[1] S. Peng, L. Zou, W. Liu, C. Liu, D.J. McClements, Fabrication and characterization of curcumin-loaded liposomes formed from sunflower lecithin: impact of composition and environmental stress, J. Agric. Food Chem. 66 (2018) 12421–12430.

2.  Some editing improvements are required. The equations are missing. The quality of the figures must be improved.

Response: As suggested,we have re-edited the equations and figures. Please see the revised manuscript with yellow highlight. Please see Lines 99, 150, 170, 231, 243-244 and 314.

3.  Line 302: change mothed with method

Response: As suggested, we have revised this word, please see Line 305.

4.  Line 331-332 what do you mean?

Response: As recommended by the reviewer, we have rewritten the sentences, please see Lines 334-336.

Reviewer 3 Report

The introduction contains very simple information, and from the basics of the research topic, it could have been abbreviated so that the reader's focus is on the new and important in this field.

A good research idea is to use chitosan compounds coated with biologically active compounds, such as curcumin-containing, that are very important to the body and use them as compounds that increase the stability of biological compounds to benefit from them to the maximum possible degree in the body and gastrointestinal digestion.
The research, from my point of view, accepts in the present form.

Author Response

  • We thank the reviewer for their positive comments.